Journal of Machine Learning Research 23 (2024) 1-12          Submitted 1/24; Revised 5/24; Published 9/24

# Multi-head Attention-based Deep Multiple Instance Learning

**Hassan Keshvarikhojasteh**               H.KESHVARIKHOJASTEH@TUE.NL
**Josien P.W. Pluim**                       J.PLUIM@TUE.NL
**Mitko Veta**                                 M.VETA@TUE.NL
*Department of Biomedical Engineering, Eindhoven University of Technology, Eindhoven, The Netherlands*

**Editor:**

## Abstract

This paper introduces MAD-MIL, a Multi-head Attention-based Deep Multiple Instance Learning model, designed for weakly supervised Whole Slide Images (WSIs) classification in digital pathology. Inspired by the multi-head attention mechanism of the Transformer, MAD-MIL simplifies model complexity while achieving competitive results against advanced models like CLAM and DS-MIL. Evaluated on the MNIST-BAGS and public datasets, including TUPAC16, TCGA BRCA, TCGA LUNG, and TCGA KIDNEY, MAD-MIL consistently outperforms ABMIL. This demonstrates enhanced information diversity, interpretability, and efficiency in slide representation. The model's effectiveness, coupled with fewer trainable parameters and lower computational complexity makes it a promising solution for automated pathology workflows. Our code is available at https://github.com/tueimage/MAD-MIL.

**Keywords:** Whole Slide Image, Multiple Instance Learning, ABMIL, MAD-MIL.

## 1 Introduction

Digital pathology leverages digital imaging technology to create high-resolution images of pathology slides, which can then be viewed, analyzed, and shared electronically. The adoption of digital pathology offers several advantages, including increased efficiency, improved collaboration, and the potential for more accurate and reproducible diagnoses Williams et al. (2017). Deep learning has emerged as a powerful tool in digital pathology, offering advanced image analysis capabilities. Convolutional Neural Networks (CNNs) Krizhevsky et al. (2012) and recently Vision Transformers (ViTs) Dosovitskiy et al. (2020) excel in learning complex patterns from large datasets and have been applied to various aspects of digital pathology Coudray et al. (2018); Litjens et al. (2017); Campanella et al. (2019). Despite the promising potential of deep learning in digital pathology, its adoption encounters various challenges Esteva et al. (2019); Holzinger et al. (2017); Madabhushi and Lee (2016). Challenges include the need for large and diverse annotated datasets, model generalization across different institutions and populations, and ethical considerations. Multiple Instance Learning (MIL) has been widely utilized in digital pathology, to address challenges associated with weakly labeled annotated datasets. In MIL, data is organized into bags, and each bag contains multiple instances, with the label of the bag determined by the presence of at least one positive instance. Attention-based deep Multiple Instance Learning (ABMIL) Ilse et al. (2018) introduces an attention mechanism that allows the model to selectively focus

Figure 1: The overall framework of the weakly supervised WSI classification task using MIL.

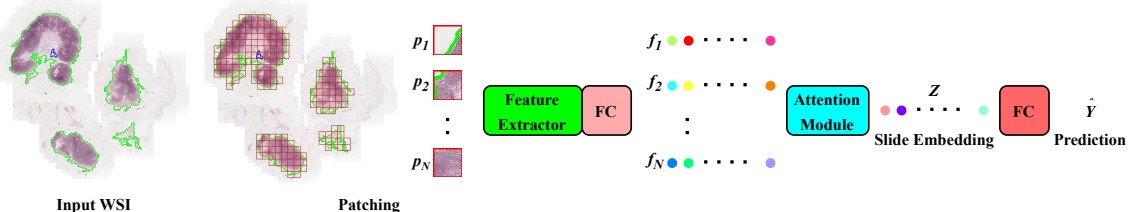

on crucial instances within each bag. By assigning different attention weights to various instances, ABMIL can effectively evaluate the importance of individual regions within a whole slide image, leading to more accurate predictions and adding a layer of interpretability. Clustering-Constrained Attention Multiple Instance Learning (CLAM) Lu et al. (2021) builds upon the foundations of ABMIL, introducing a novel clustering layer to refine the feature space and adopting it for multi-class weakly supervised WSI classification. Dual-Stream Multiple Instance Learning Network (DS-MIL) Li et al. (2021) develops a novel MIL aggregator in a dual-stream architecture outperforming previous methods. Recently, the Attention-Challenging MIL (ACMIL) Zhang et al. (2023) has witnessed notable advancements using two key techniques: Multiple Branch Attention (MBA) and Stochastic TopK Instance Masking (STKIM). They have enabled ACMIL to enhance its discriminative instance capturing capabilities and mitigate the impact of prominent top-k instances, respectively.

One of the key components of the Transformer architecture Vaswani et al. (2017) is the multi-head self-attention mechanism, which runs the attention mechanism in parallel across multiple heads to capture diverse aspects of relationships. Although recent models advanced the state of the art for weakly supervised WSIs classification, they increased the complexity of the ABMIL model with respect to architecture design and number of trainable parameters. Aiming for simpler models, we propose the Multi-head ABMIL (MAD-MIL) model for weakly supervised WSIs classification. It is inspired by the multi-head self-attention mechanism not only to reduce the number of trainable parameters but also to incorporate various facets of the input WSI using multiple attention heads. Although there is a multi-head version of ABMIL available in the official repository, our method differs from that implementation. In ABMIL, attention weights are computed using full-dimensional input features, and a larger classifier is employed for the larger slide representation. Consequently, this approach increases the number of parameters and the overall complexity of ABMIL. While similar ideas have been explored in previous works such as Li et al. (2019) for multi-tagging WSIs and ACMIL for weakly supervised WSI classification, our model distinguishes itself from Li et al. (2019) in two key ways. First, we adhere to the original architecture proposed in ABMIL to maintain simplicity in the developed model. Second, the MAD-MIL is specifically designed for the task of weakly supervised WSIs classification while the model proposed by Li et al. (2019) was designed for multi-tagging WSIs using the Multi-Tag Attention Module to construct tag-related global slide representations for final

Figure 2: The attention component used in ABMIL(Left), where only one attention module is utilized, and in MAD-MIL(Right), where multiple attention modules are incorporated.

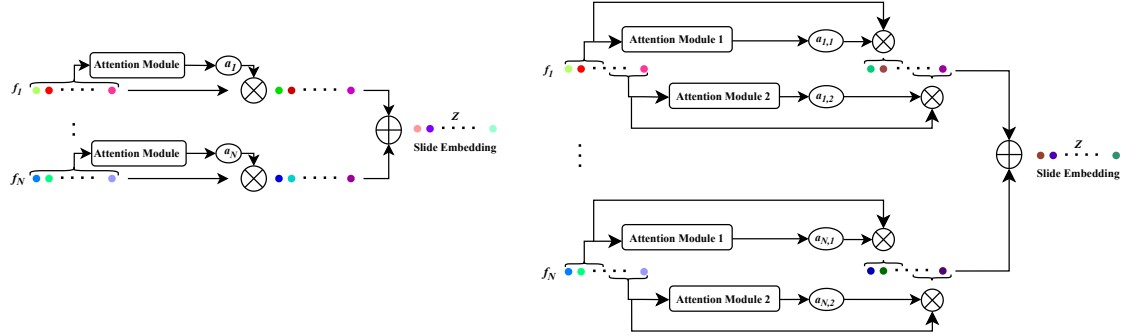

prediction. Regarding ACMIL, several attention weights are computed for input features alongside the introduction of two supplementary regularization terms: semantic and diversity regularizations. In contrast to ACMIL, our approach within ABMIL does not entail the extraction of attention weights utilizing full-dimensional input features. This choice is made to avoid the addition of extra parameters. Furthermore, we abstain from introducing additional regularization terms, thereby preserving the simplicity of the ABMIL framework. We evaluate MAD-MIL for weakly supervised classification both in controlled settings, using variants of the MNIST-BAGS dataset, and on four publicly available clinical datasets, namely, TUPAC16, TCGA BRCA, TCGA LUNG, and TCGA KIDNEY.

## 2 Method

### 2.1 Problem formulation

If we consider the input WSI as $I$, then our goal is to predict its slide-level label $Y$. As it is infeasible to feed the $I$ directly to the network due to computational constraints, it is tiled into $N$ small instances $I = \{p_1, p_2, \ldots, p_N\}$ to create a bag of instances, with unknown labels. To predict the slide label, in the first step the instances' features are extracted using a pre-trained network and compressed by a fully connected layer $\{f_1, f_2, \ldots, f_N\} = F(pre-trained\{p_1, p_2, \ldots, p_N\}), \forall n \, f_n \in \mathbb{R}^{1 \times D}$. In the second step, the features are aggregated with an attention-based function to obtain the slide embedding $Z = H(f_1, f_2, \ldots, f_N), Z \in \mathbb{R}^{1 \times D}$. Finally, the slide-label is produced with a fully connected layer $\hat{Y} = L(Z)$. Figure 1 shows the overall framework of the approach.

### 2.2 Attention-based deep MIL

In this method, two types of bags (positive and negative) are considered for binary WSIs classification. Positive bags contain at least one key-instance, while there are none in negative bags. Then, two types of attention functions are proposed, which we only present the gated version for brevity Ilse et al. (2018):

Table 1: The average test AUC and F1 scores (±std) for MNIST-BAGS. The flops are measured with 120 instances per bag.

| Model | AUC | F1 | Model Size | FLOPs |
|-------|-----|-----|-----------|-------|
| p-pos= 0.4, p-neg= 0.2 | | | | |
| ABMIL | 0.803 ± 0.059 | 0.683 ± 0.120 | 167.1 K | 19.9 M |
| **MAD-MIL/6** | **0.845 ± 0.032** | **0.750 ± 0.032** | **107.1 K** | **12.7 M** |
| Mean-Pool | 0.860 ± 0.021 | 0.774 ± 0.028 | 100.8 K | 12.0 M |
| Max-Pool | 0.723 ± 0.034 | 0.653 ± 0.035 | 100.8 K | 12.0 M |
| p-pos= 0.6, p-neg= 0.4 | | | | |
| ABMIL | 0.831 ± 0.016 | 0.727 ± 0.021 | 167.1 K | 19.9 M |
| **MAD-MIL/4** | **0.840 ± 0.019** | **0.740 ± 0.015** | **105.1 K** | **12.5 M** |
| Mean-Pool | 0.864 ± 0.021 | 0.774 ± 0.022 | 100.8 K | 12.0 M |
| Max-Pool | 0.713 ± 0.042 | 0.646 ± 0.042 | 100.8 K | 12.0 M |
| p-pos= 0.8, p-neg= 0.6 | | | | |
| ABMIL | 0.80 ± 0.057 | 0.668 ± 0.104 | 167.1 K | 19.9 M |
| **MAD-MIL/7** | **0.835 ± 0.026** | **0.753 ± 0.028** | **107.2 K** | **12.8 M** |
| Mean-Pool | 0.870 ± 0.026 | 0.783 ± 0.039 | 100.8 K | 12.0 M |
| Max-Pool | 0.784 ± 0.024 | 0.698 ± 0.051 | 100.8 K | 12.0 M |

$$a_n = \frac{exp\{\mathbf{w}^\top(tanh(\mathbf{V}f_n{}^\top) \odot sigm(\mathbf{U}f_n{}^\top))\}}{\sum_{l=1}^{N} exp\{\mathbf{w}^\top(tanh(\mathbf{V}f_l{}^\top) \odot sigm(\mathbf{U}f_l{}^\top))\}} \tag{1}$$

With $\mathbf{U}, \mathbf{V} \in \mathbb{R}^{d \times D}$ and $\mathbf{w} \in \mathbb{R}^{d \times 1}$ learnable parameters, $\odot$ an element-wise multiplication and $sigm()$ the sigmoid function. The attention weights $a_n$ ideally assign higher values to key-instances and lower values to non-key ones, providing interpretability of the model.

## 2.3 Multi-head attention-based deep MIL

Empirical evidence suggests that the multi-head self-attention mechanism in the Transformer outperforms its single-head counterpart Vaswani et al. (2017). Additionally, by leveraging the attention weights from each head, we can generate diverse heatmaps that enhance the interpretability of the network. Since our goal is to employ the attention function from ABMIL, which differs significantly from the scaled dot-product self-attention module in the Transformer, a direct adaptation of the multi-head idea is impractical. Therefore, we propose to divide the input features into $M$ equal parts, i.e. $\forall n, m \; f_{n,m} \in \mathbb{R}^{1 \times \lceil \frac{D}{M} \rceil}; n = \{1 : N\}, m = \{1 : M\}$. Then, we feed each part to distinct attention modules given by:

$$a_{n,m} = \frac{exp\{\mathbf{w_m}^\top(tanh(\mathbf{V_m}f_{n,m}{}^\top) \odot sigm(\mathbf{U_m}f_{n,m}{}^\top))\}}{\sum_{l=1}^{N} exp\{\mathbf{w_m}^\top(tanh(\mathbf{V_m}f_{l,m}{}^\top) \odot sigm(\mathbf{U_m}f_{l,m}{}^\top))\}} \tag{2}$$

In the next step, the features of each part are aggregated and ultimately, slide representation is calculated as the concatenation of the aggregated features:

Table 2: Slide-Level Classification Evaluation on TUPAC16. The flops are measured with 120 instances per bag, and the instance feature extraction by ResNet-50 is not considered in the presented model sizes and flops.

| Model | AUC | F1 | Model Size | FLOPs |
|-------|-----|-----|------------|-------|
| ABMIL | $0.790 \pm 0.013$ | $0.725 \pm 0.013$ | 788.7 K | 94.4 M |
| **MAD-MIL/3** | $0.802 \pm 0.006$ | $\mathbf{0.735 \pm 0.009}$ | **614.8 K** | **73.5 M** |
| CLAM-MB | $0.803 \pm 0.008$ | $0.725 \pm 0.010$ | 791.0 K | 94.4 M |
| CLAM-SB | $0.796 \pm 0.004$ | $0.730 \pm 0.007$ | 790.7 K | 94.4 M |
| DS-MIL | $0.796 \pm 0.011$ | $0.724 \pm 0.008$ | 1186.9 K | 142.0 M |
| **ACMIL** | $\mathbf{0.803 \pm 0.004}$ | $0.729 \pm 0.008$ | 794.8 K | 94.5 M |
| Mean-Pool | $0.790 \pm 0.004$ | $0.707 \pm 0.012$ | 525.8 K | 62.9 M |
| Max-Pool | $0.799 \pm 0.003$ | $0.719 \pm 0.010$ | 525.8 K | 62.9 M |

$$\hat{Z} = Concat(\sum_{n=1}^{N} a_{n,1} f_{n,1}, \ldots, \sum_{n=1}^{N} a_{n,M} f_{n,M}) \tag{3}$$

The slide label can be predicted same as before $\acute{Y} = L(\hat{Z})$. Our developed method (MAD-MIL) has fewer trainable parameters, as we use smaller attention modules and it can return various attention heatmaps. However, one more hyperparameter M (number-of-heads) should be tuned. Figure 2 presents ABMIL and MAD-MIL models.

## 3 Experiment and Results

### 3.1 Dataset

We used four public WSI datasets. The **TUPAC16** dataset Veta et al. (2019) comprises 821 H&E slides from the Cancer Genome Atlas (TCGA) project, annotated with breast tumor proliferation labels. We categorize the labels into two classes: low-grade (slides labeled as grade 1) and high-grade (slides labeled as grade 2 or 3). The **TCGA Breast Invasive Carcinoma Cancer (TCGA BRCA)** dataset includes 1038 WSIs for two-types of breast cancer; Invasive Ductal Carcinoma (IDC) versus Invasive Lobular Carcinoma (ILC) with slide-level labels. The **TCGA Lung Cancer** dataset is a public dataset for Non-Small Cell Lung Carcinoma (NSCLC) subtyping. The dataset contains 1046 slides for Lung Adenocarcinoma (LUAD) and Lung Squamous Cell Carcinoma (LUSC) with only slide-level labels. The **TCGA Kidney Cancer** includes 918 slides for three types of kidney cancer, namely Clear Cell Renal Cell Carcinoma (CCRCC), Papillary Renal Cell Carcinoma (PRCC), and Chromophobe Renal Cell Carcinoma (CHRCC).

We also adopt the approach proposed in Ilse et al. (2018) to construct a dataset by utilizing images from the MNIST image dataset. In practical applications, particularly in weakly supervised WSI classification, a fundamental assumption of conventional MIL is challenged, as negative bags may also encompass some key-instances Li and Vasconcelos

| Model | AUC | F1 | Model Size | FLOPs |
|---|---|---|---|---|
| ABMIL | $0.882 \pm 0.046$ | $0.783 \pm 0.061$ | 788.7 K | 94.4 M |
| **MAD-MIL/2** | $0.897 \pm 0.058$ | $0.791 \pm 0.064$ | **657.6 K** | **78.6 M** |
| **CLAM-MB** | $0.897 \pm 0.052$ | $\mathbf{0.809 \pm 0.060}$ | 791.0 K | 94.4 M |
| CLAM-SB | $0.889 \pm 0.046$ | $0.790 \pm 0.052$ | 790.7 K | 94.4 M |
| DS-MIL | $0.903 \pm 0.053$ | $0.788 \pm 0.056$ | 1186.9 K | 142.0 M |
| **ACMIL** | $\mathbf{0.905 \pm 0.039}$ | $0.793 \pm 0.054$ | 794.8 K | 94.5 M |
| Mean-Pool | $0.891 \pm 0.057$ | $0.793 \pm 0.060$ | 525.8 K | 62.9 M |
| Max-Pool | $0.900 \pm 0.056$ | $0.795 \pm 0.050$ | 525.8 K | 62.9 M |

Table 3: Slide-Level Classification Evaluation on TCGA BRCA dataset.

(2015). To address this, we introduce soft-bags and designate the digit 8 as the key-instance and create positive and negative bags with varying numbers of the key-instance.

### 3.2 Implementation Details

For the MNIST-BAGS dataset, we create 50 bags for training, 100 for evaluation, and 900 for testing, every bag containing 20 images. In our experimental setup, we explore varying percentages of the key-instance within positive and negative bags (p-pos, p-neg.) Specifically, in our analysis, we consider the (0.4, 0.2), (0.6, 0.4), and (0.8, 0.6) pairs. These pairs correspond to positive bags with values 0.4, 0.6, 0.8 and negative bags with values 0.2, 0.4, 0.6. We directly input MNIST images into the model, $(input\_dim = 28 \times 28 = 784$ pixels), and then they are compressed to a size of $D = 128$. Throughout these experiments, we utilize the Adam optimizer and train models for 20 epochs and identify optimal learning rate and weight decay values based on the validation loss. We repeat the experiments with ten different seeds and report the mean AUC and F1 values to maintain consistency with prior methodologies. For the preprocessing of the TUPAC16 and TCGA datasets, we adhere to the steps outlined in Lu et al. (2021), incorporating the extraction of features from non-overlapping 256×256 patches at 20× magnification using the ImageNet pretrained ResNet50. We feed extracted features into the model, where the feature dimensionality is $input\_dim = 1024$. These features are then compressed to a size of $D = 512$. We repeat the experiments five times to evaluate our framework on the TUPAC16 dataset, as there is an official test split. We report the average performance of the test set for the five experiments. However, for the TCGA datasets we employ the same 10-folder cross-validation setup as adopted in Chen et al. (2022), and present the mean and standard variance values of AUC and F1 scores. The training of the models for these datasets involves using the Adam optimizer for 50 epochs. A learning rate of 1e-4 is employed, and we tune the weight-decay value based on the validation loss.

For all experiments we determine the optimal number of heads (M) according to the validation loss value. Regarding the Mean Pool and Max Pool models, we ensure a fair comparison by implementing the bag embedding-based variant as opposed to the instance-level. Specifically, after the feature compression step, we calculate the slide representation using either the max or mean operation and subsequently predict the slide label. In what

| Model | AUC | F1 | Model Size | FLOPs |
|---|---|---|---|---|
| ABMIL | 0.931 ± 0.020 | 0.853 ± 0.034 | 788.7 K | 94.4 M |
| **MAD-MIL/8** | **0.940 ± 0.015** | **0.872 ± 0.027** | **559.3 K** | **66.8 M** |
| CLAM-MB | 0.935 ± 0.017 | 0.868 ± 0.039 | 791.0 K | 94.4 M |
| CLAM-SB | 0.932 ± 0.015 | 0.868 ± 0.034 | 790.7 K | 94.4 M |
| DS-MIL | 0.937 ± 0.015 | 0.868 ± 0.032 | 1186.9 K | 142.0 M |
| ACMIL | 0.940 ± 0.020 | 0.871 ± 0.041 | 794.8 K | 94.5 M |
| Mean-Pool | 0.905 ± 0.021 | 0.844 ± 0.028 | 525.8 K | 62.9 M |
| Max-Pool | 0.950 ± 0.013 | 0.867 ± 0.035 | 525.8 K | 62.9 M |

Table 4: Slide-Level Classification Evaluation on TCGA LUNG dataset.

follows we use brief notation to indicate the number of heads for the MAD-MIL: for instance, MAD-MIL/3 means the MAD-MIL model with three heads.

### 3.3 Quantitative Results

Table 1 reports the outcomes obtained from the MNIST-BAGS. Across all three configurations, the MAD-MIL model consistently exhibits superior performance compared to AB-MIL, despite having a lower number of trainable parameters and computational complexity. Notably, the most substantial enhancements are observed in the last setting (p-pos= 0.8, p-neg= 0.6), where AUC and F1 metrics experience notable increases of 4.3% and 12.7%, respectively. The results for the weakly-supervised WSI classification are presented in Tables 2, 3, 4 and 5. Notably, the MAD-MIL model outperforms ABMIL across all datasets. Interestingly, it achieves comparable performance to sophisticated networks like CLAM and DS-MIL. It is worth noting that while the Mean-Pool and Max-Pool models exhibit similar performance to other models, their lack of interpretability hinders their practical application.

### 3.4 Qualitative Results

We present attention heatmaps in Figure 4 to assess the interpretability of the methods applied to a LUAD slide from the TCGA LUNG dataset. Notably, all three methods (ABMIL, CLAM-MB, and MAD-MIL/8) focus on similar regions within the input WSI, demonstrating negligible differences. However, MAD-MIL distinguishes itself by generating additional heatmaps. This enhancement could potentially contribute to the overall interpretability of the network.

## 4  Conclusion

In summary, this paper presents the MAD-MIL model for weakly supervised WSIs classification in digital pathology. By leveraging the multi-head attention mechanism inspired by the Transformer architecture, MAD-MIL captures diverse aspects of input WSIs. Evaluation on MNIST-BAGS, TUPAC16 and TCGA datasets demonstrates consistent outperformance compared to the ABMIL and comparable results to advanced models like CLAM and DS-MIL. The integration of multiple attention heads not only enriches information diversity in

| Model | AUC | F1 | Model Size | FLOPs |
|---|---|---|---|---|
| ABMIL | 0.983 ± 0.010 | 0.894 ± 0.037 | 788.7 K | 94.4 M |
| **MAD-MIL/5** | **0.985 ± 0.007** | 0.898 ± 0.034 | **582.7 K** | **69.6 M** |
| CLAM-MB | 0.982 ± 0.009 | 0.893 ± 0.043 | 791.0 K | 94.4 M |
| CLAM-SB | 0.983 ± 0.008 | 0.889 ± 0.026 | 790.7 K | 94.4 M |
| DS-MIL | 0.983 ± 0.009 | **0.908 ± 0.037** | 1186.9 K | 142.0 M |
| ACMIL | 0.984 ± 0.008 | 0.907 ± 0.027 | 794.8 K | 94.5 M |
| Mean-Pool | 0.978 ± 0.009 | 0.880 ± 0.019 | 525.8 K | 62.9 M |
| Max-Pool | 0.989 ± 0.006 | 0.914 ± 0.031 | 525.8 K | 62.9 M |

Table 5: Slide-Level Classification Evaluation on TCGA KIDNEY dataset.

slide representation but also enhances interpretability, showcasing efficiency with a reduced number of trainable parameters.

## Acknowledgments and Disclosure of Funding

We thank Ruben T. Lucassen for valuable discussions and feedback. This work was done as a part of the IMI BigPicture project (IMI945358).

Figure 3: Attention heatmaps generated by different models for a LUAD slide selected from the TCGA LUNG dataset. Top row: from left to right, the attention heatmap produced by the ABMIL and CLAM-MB. Bottom rows: from top to bottom and left to right, the attention heatmap generated by the MAD-MIL/8-head-1:8.

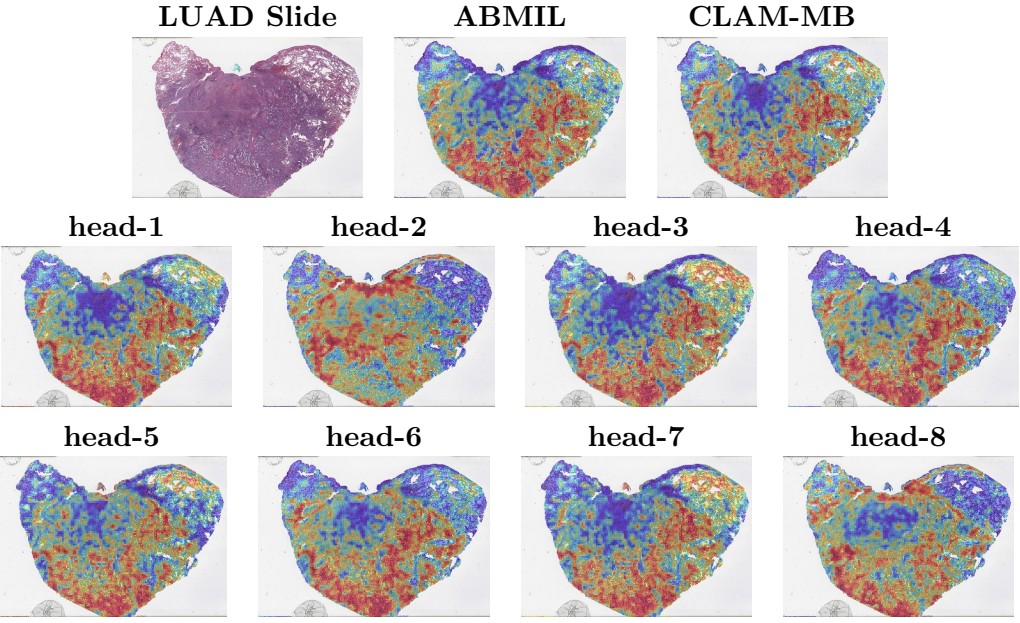

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

Figure 4: Attention weights generated by ABMIL and MAD-MIL/6 networks for a negative and postive bag selected from the MNIST-BAGS {p-pos= 0.4, p-neg= 0.2} dataset. For each digit the corresponding attention weight is given by the trained network. Top row: from top to bottom , the attention weights produced by the ABMIL and MAD-MIL/6-head-1:6 for a negative bag. Bottom row: from top to bottom, the attention weights produced by the ABMIL and MAD-MIL/6-head-1:6 for a positive bag.

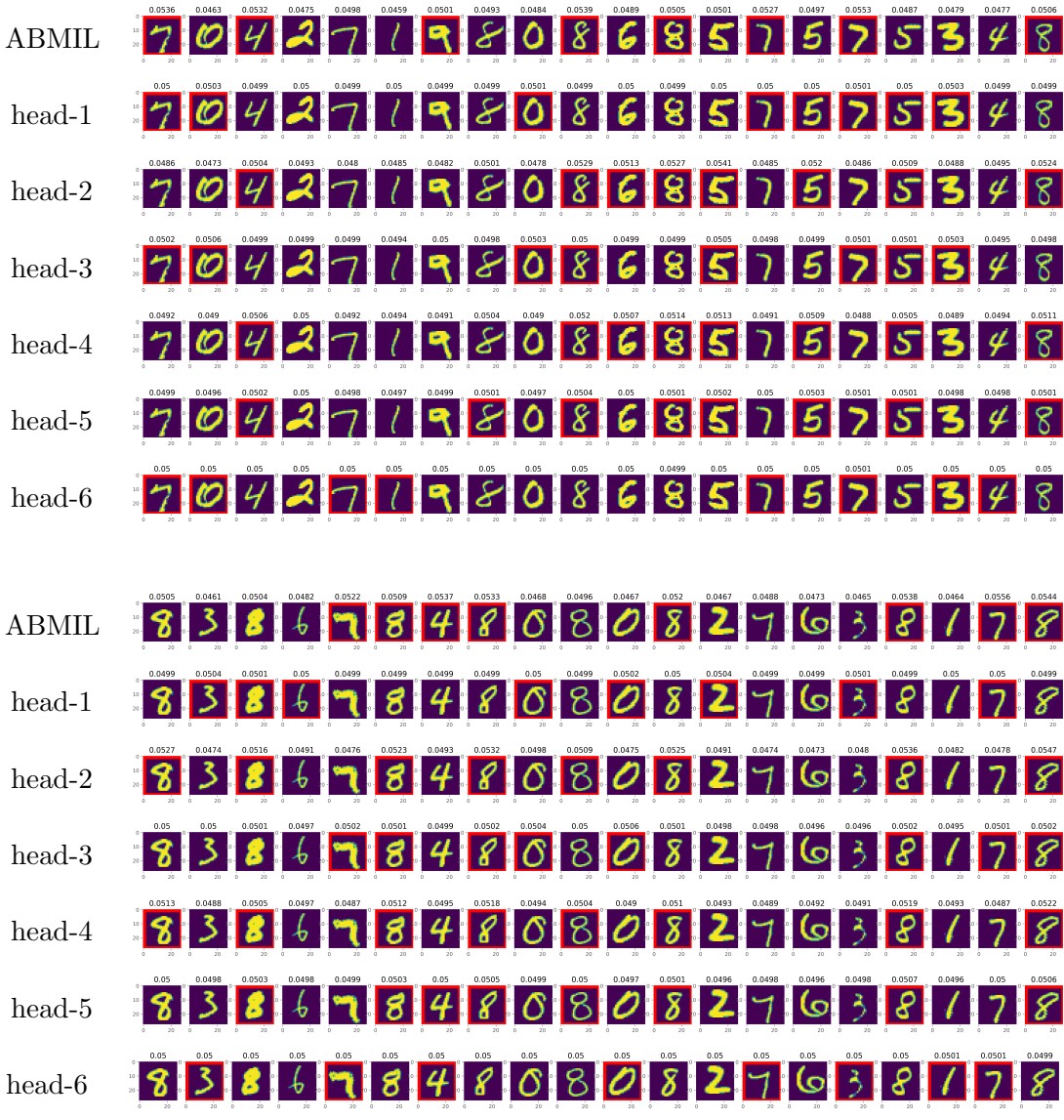

## 5 Supplementary Material

### 5.1 Qualitative Results on MNIST-BAGS

We present the attention weights of the ABMIL and MAD-MIL models on the MNIST-BAGS dataset in Figure 4 to delve deeper into their interpretability. Unlike ABMIL, which distributes attention across both key and non-key instances, certain heads of the MAD-MIL model focus solely on either key instances (e.g., head-2 and head-4) or non-key instances (e.g., head-1 and head-6).

### 5.2 Ablation Study

Figure 5 presents the AUC, ACC, and Val Loss of the proposed method with respect to various number of heads on the datasets. These curves indicate that the multi-head idea is beneficial to both AUC and ACC metrics. However, ACC is more sensitive to the number of heads, with a significant drop observed for certain head counts. Conversely, AUC tends to consistently improve with an increase in the number of heads. Looking ahead, MAD-MIL opens avenues for further research and exploration. Future work could involve adapting the model for specific pathology tasks or refining its applicability across diverse clinical settings.

Figure 5: The impact of the number of heads (M) on the average AUC, ACC and Val Loss. Each column presents the results of one dataset, from left to right: TUPAC16, TCGA BRCA, TCGA LUNG, and TCGA KIDNEY.

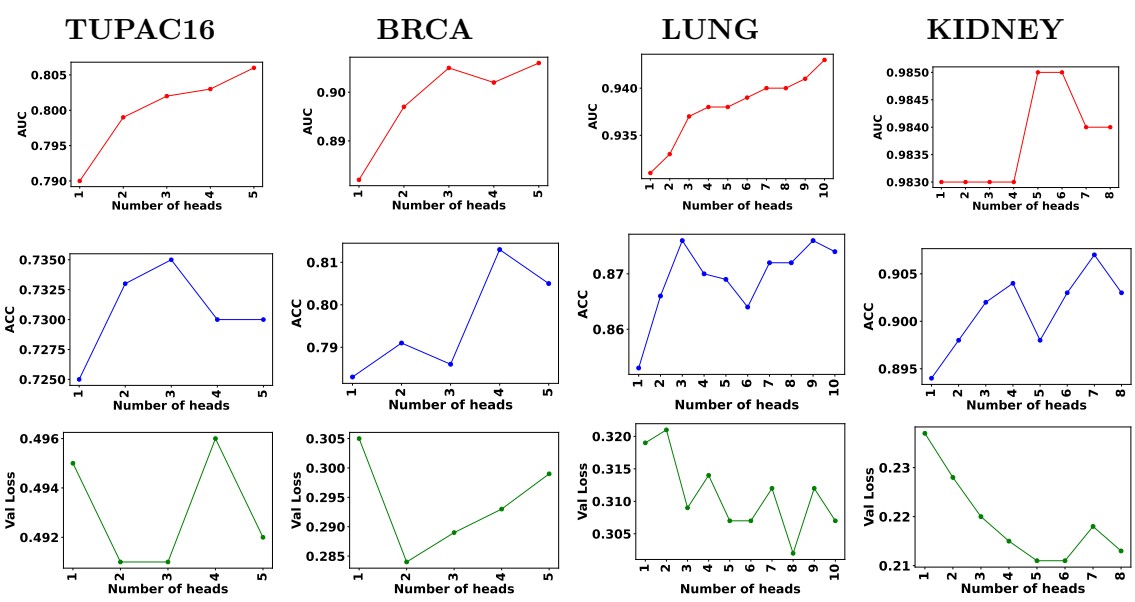

