# OpenReview forum: "Multi-head Attention-based Deep Multiple Instance Learning"
_MICCAI.org/2024/Workshop/COMPAYL — COMPAYL 2024_

### Official Review · Reviewer_3CWP · 2024-07-09
**Review of paper 1**

**Custom Rating:** 4
**Confidence:** 4

**Review:**

The paper extends ABMIL within the context of the slide-level classification task in pathology, introducing multi-head attention to map and aggregate patch-level embeddings into slide-level embeddings. The paper extensively validates the approach on real pathology slide datasets from TUPAC16, TCGA-BRCA, TCGA-Lung, and TCGA-Kidney, demonstrating the superiority of using multi-head attention. However, there are some details that need addressing.

1. Which pre-trained network was used as the feature extractor? This should be specified.

2. Figure 5 visualizes the impact of the number of heads on performance. To facilitate comparison with the baseline (a single head), the result for "1" head should be added to the figure. Additionally, why are the experiment head counts not consistent across different experiments (e.g., some range from 2-5, while others from 2-10)? This should be standardized.

3. What do p-pos and p-neg represent? This should be clarified.

4. Moreover, the article appears to group based on the order of channels. I would be curious to know how much performance fluctuation occurs if the channel order is shuffled.

---

### Official Review · Reviewer_nxkM · 2024-07-09
**Next step of ABMIL with some efficiency gains.**

**Custom Rating:** 4
**Confidence:** 4

**Review:**

The paper outlines the multihead attention formulation of the WSI MIL problem. Compared to ACMIL, the efficiency gains appear modest. I would love to see a more real-world metric that captures the performance / compute (FLOPs) tradeoff that demonstrates the clear superiority of the proposed method.

As the paper stands, the multihead formulation makes sense. The metrics look good. The new method is a good reference for people looking to develop attention based MIL models.

---

### Official Review · Reviewer_1ijR · 2024-07-15
**A lighter multi-head attention-based MIL framework for pathology application**

**Custom Rating:** 4
**Confidence:** 4

**Review:**

The authors present a new lighter (in terms of number of parameters) multi-head attention-based MIL algorithm and evaluate it on several pathology datasets (TUPAC16, and several TCGA datasets) as well as bags created from MNIST data. The paper is well written and the results indicate that the proposed method works well for pathology applications. However, the discussion could be expanded as currently many results are presented in the tables and not properly discussed. For example, ACMIL method (which also uses multi-head approach) performs on part or better than the proposed method, hence this could have been discussed in more detail and suggestions made why the reader could still prefer their proposed method.

The experiments on MNIST data with varying fraction of important instances in positive and negative bags is very interesting. It would be interesting to replicate this experiment with pathology data to see if the results are similar. Of course, this would require to have detailed labels which can be expensive to do. Finally, some details of implementation are missing such as what network was used to extract features, i.e. what data was it trained on.

---

### Decision · Program_Chairs · 2024-07-16

Accept